# Targeted Delivery of BZLF1 to DEC205 Drives EBV-Protective Immunity in a Spontaneous Model of EBV-Driven Lymphoproliferative Disease

**DOI:** 10.3390/vaccines9060555

**Published:** 2021-05-26

**Authors:** Elshafa Hassan Ahmed, Eric Brooks, Shelby Sloan, Sarah Schlotter, Frankie Jeney, Claire Hale, Charlene Mao, Xiaoli Zhang, Eric McLaughlin, Polina Shindiapina, Salma Shire, Manjusri Das, Alexander Prouty, Gerard Lozanski, Admasu T. Mamuye, Tamrat Abebe, Lapo Alinari, Michael A. Caligiuri, Robert A. Baiocchi

**Affiliations:** 1Department of Veterinary Biosciences, College of Veterinary Medicine, The Ohio State University, Columbus, OH 43210, USA; elshafa.ahmed@osumc.edu (E.H.A.); Shelby.Sloan@osumc.edu (S.S.); 2Comprehensive Cancer Center, The James Cancer Hospital and Solove Research Institute, The Ohio State University, Columbus, OH 43210, USA; eric.brooks@osumc.edu (E.B.); Sarah.schlotter@osumc.edu (S.S.); frankiej336@gmail.com (F.J.); Charlene.mao@osumc.edu (C.M.); Polina.shindiapina@osumc.edu (P.S.); dasmanjusri7@gmail.com (M.D.); Ap2794@mynsu.nova.edu (A.P.); Lapo.Alinari@osumc.edu (L.A.); 3Department of Biomedical Engineering, College of Engineering, The Ohio State University, Columbus, OH 43210, USA; claire.hale1@outlook.com; 4Department of Biomedical Informatics/Center for Biostatistics, The Ohio State University, Columbus, OH 43210, USA; Xiaoli.zhang@osumc.edu (X.Z.); eric.mclaughlin@osumc.edu (E.M.); 5Division of Hematology, Department of Internal Medicine, College of Medicine, The Ohio State University, Columbus, OH 43210, USA; 6College of Education and Human Ecology, The Ohio State University, Columbus, OH 43210, USA; Salma.shire@case.edu; 7Department of Pathology, The Ohio State University, Columbus, OH 43210, USA; Gerard.lozanski@osumc.edu; 8Department of Internal Medicine, Black Lion Hospital, Addis Ababa University, Addis Ababa 3614, Ethiopia; kadmasen@gmail.com; 9Department of Microbiology, Black Lion Hospital, Addis Ababa University, Addis Ababa 3614, Ethiopia; tamrat.abebe@aau.edu.et; 10City of Hope National Medical Center, Duarte, CA 91010, USA

**Keywords:** Epstein-Barr virus, BZLF1, BZLF1-specific cytotoxic T-cells, vaccine, post-transplant lymphoproliferative disease (PTLD), Hu-PBL-SCID model

## Abstract

Epstein-Barr virus (EBV) is a human herpes virus that infects over 90% of the world’s population and is linked to development of cancer. In immune-competent individuals, EBV infection is mitigated by a highly efficient virus-specific memory T-cell response. Risk of EBV-driven cancers increases with immune suppression (IS). EBV-seronegative recipients of solid organ transplants are at high risk of developing post-transplant lymphoproliferative disease (PTLD) due to iatrogenic IS. While reducing the level of IS may improve EBV-specific immunity and regression of PTLD, patients are at high risk for allograft rejection and need for immune-chemotherapy. Strategies to prevent PTLD in this vulnerable patient population represents an unmet need. We have previously shown that BZLF1-specific cytotoxic T-cell (CTL) expansion following reduced IS correlated with immune-mediated PTLD regression and improved patient survival. We have developed a vaccine to bolster EBV-specific immunity to the BZLF1 protein and show that co-culture of dendritic cells (DCs) loaded with a αDEC205-BZLF1 fusion protein with peripheral blood mononuclear cells (PMBCs) leads to expansion and increased cytotoxic activity of central-effector memory CTLs against EBV-transformed B-cells. Human–murine chimeric Hu-PBL-SCID mice were vaccinated with DCs loaded with αDEC205-BZLF1 or control to assess prevention of fatal human EBV lymphoproliferative disease. Despite a profoundly immunosuppressive environment, vaccination with αDEC205-BZLF1 stimulated clonal expansion of antigen-specific T-cells that produced abundant IFNγ and significantly prolonged survival. These results support preclinical and clinical development of vaccine approaches using BZLF1 as an immunogen to harness adaptive cellular responses and prevent PTLD in vulnerable patient populations.

## 1. Introduction

The Epstein-Barr virus (EBV) is a gamma herpes virus [1] that infects over 90% of the adult population worldwide [2]. EBV is trophic to human naïve B-cells and upon primary infection is capable of driving cellular immortalization and transformation. In immunocompetent individuals, immortalized B-cells are controlled by a highly efficient, EBV-specific, memory helper and cytotoxic T-lymphocyte (CTL) response [3]. Although this immune response can control the virus during primary infection, EBV adopts a latency phase and establishes life-long persistence in the human memory B-cell compartment. The virus is capable of entering a lytic cycle program where viral replication and assembly of infectious virions can occur, placing immunocompromised individuals at risk for a variety of lymphoproliferative disorders (EBV-LPD) [4,5,6]. Solid organ transplant recipients treated with immunosuppressive medications to prevent graft rejection are at particularly high risk for developing a form of EBV-LPD known as post-transplant lymphoproliferative disease (PTLD) [7]. Iatrogenic immune suppression (IS) impairs T-cell immunity, allowing EBV to escape immune surveillance and drive B-cell immortalization and transformation [8]. EBV-seronegative transplant candidates are at the greatest risk for developing primary EBV infection and PTLD [9,10].

PTLD usually occurs during the first year following transplantation [11] and can range from a polyclonal, infectious mononucleosis type syndrome to high-grade, monomorphic diffuse large B-cell lymphoma associated with poor outcome [7,12,13]. Current clinical guidelines for treatment of PTLD recommend stepwise, individualized treatment plans based on the timing of onset, subtype of disease, and response to initial treatment; however, there is no universally approved treatment regimen [14,15]. While reduction in IS and treatment with the CD20-specific monoclonal antibody, rituximab, may lead to immune-reconstitution and resolution of PTLD, the risk of graft rejection is unacceptably high [16]. Upfront management with reduced IS and single-agent rituximab has led to durable remissions; however, prognosis remains poor in rituximab non-responders [17].

Infusion of EBV-specific cytotoxic T-cells (CTL) can effectively treat EBV-associated diseases such as PTLD, Hodgkin’s lymphoma (HL), and nasopharyngeal carcinoma (NPC) [18,19]. Over 75% of transplant recipients with PTLD achieved a complete remission when treated with EBV-specific CTLs [20,21]. Additionally, administration of adoptive EBV-specific CTLs prevented PTLD in pediatric hematopoietic stem cell transplant recipients [22]. While the use of adoptive cell therapy to prevent and treat PTLD highlights the importance of EBV-specific cellular immune responses, cellular immunotherapy is expensive and available only at select centers with sufficient expertise [23].

Currently, there is no vaccine available to reduce the risk of PTLD [24,25]. We previously reported that patients with PTLD who showed spontaneous disease regression with reduction in IS were found to mount a robust CD3+CD8+ CTL memory response to the EBV immediate-early protein BZLF1 (BamHI Z fragment Leftward ORF1, also known as Zta) [26]. This led us to hypothesize that BZLF1 may serve as an effective immunogen to drive the expansion of EBV-specific CTL memory T-cell responses and reduce the risk of developing EBV-LPD. We chose BZLF1 as a vaccine immunogen due to the following considerations: (1) BZLF1 has been documented as the first antigen recognized by the human immune response upon primary infection [27]; (2) specific immune responses to BZLF1 correlate with spontaneous PTLD regression and better survival [26,28]; and (3) BZLF1 is essential to drive B-cell immortalization and vital to the transforming capacity of EBV and development of EBV-LPD [29].

In order to introduce BZLF1 to antigen-presenting cells, we applied a platform where BZLF1 could be delivered directly to dendritic cells (DC) expressing the endocytic receptor DEC205. We observed significant expansion of EBV-specific memory T-cell subsets capable of cytotoxic activity against autologous EBV+ lymphoblastoid cell lines (LCL). Furthermore, vaccination of an aggressive, spontaneous, in vivo model of EBV-LPD led to significant delay or prevention of EBV-LPD.

## 2. Materials and Methods

### 2.1. Study Volunteers

The Ohio State University (OSU) Office of Responsible Research Practices and Institutional Review Board (IRB) approved the study. All healthy volunteers enrolled in this study provided informed consent using our IRB-approved protocol (1998H0240). Study volunteers were screened for IgG antibody against EBV-viral capsid antigen (VCA) to confirm prior exposure to EBV.

### 2.2. Sample Collection and Storage

Peripheral blood mononuclear cells (PBMCs) from EBV+ donors were tissue-typed and HLA-B8 donors were leukapheresed to collect a large number of PBMCs for experiments. PBMCs from study volunteers were preserved in freezing media (10% dimethyl sulfoxide in fetal bovine serum (FBS)) and stored in cryopreservation tanks. EBV-transformed lymphoblastoid cell lines (EBV-LCLs) were generated from donor PBMCs as described previously [30,31] and are detailed in Appendix A.

### 2.3. Cloning and Production of Anti-DEC205-BZLF1 Vaccine

To enhance delivery of BZLF1 to antigen-presenting cells, the full-length BZLF1 open reading frame (ORF) was fused to an anti-DEC205 antibody variable region in collaboration with Celldex (Celldex Therapeutics, Hampton, NJ, USA). DEC205 is a type I cell surface protein expressed primarily by DCs [32,33]. The αDEC205-BZLF1 fusion construct was cloned into the B11 expression vector as described previously [34]. Briefly, BZLF1 cDNA was generated from the BZLF1 ORF of B95.8 strain of EBV and then inserted at the 3′ end of the B11 antibody heavy chain gene-constant region. The heavy and light chain variable regions of the anti-human DEC205 monoclonal antibody (clone 3G9) were used to replace the variable regions of the B11 antibody. The αDEC205-BZLF1 fusion protein was expressed in Chinese hamster ovary cells with the modified B11 expression vector and purified by protein A chromatography. Human chorionic gonadotropin (HCG) was utilized as a negative control and the beta chain of HCG was fused with the human αDEC205 antibody to generate αDEC205-HCG (αDEC205-Ctrl) fusion protein.

### 2.4. Dendritic Cells and PBMC Co-Culture

PBMCs from EBV+ healthy HLA-B8 donors were used for the generation of dendritic cells (DCs). A detailed description of DC generation is provided in the Appendix A. Mature, antigen-loaded DCs were co-cultured (CoCx) with autologous donor PBMCs at a ratio of 1:20 (DC: PBMC) and maintained in CTL media (43.5% *v*/*v* RPMI, 43.5% *v*/*v* Click’s medium (Sigma-Aldrich, Missouri, USA), 10% FBS, 2 mM GlutaMax, 50 uM 2-ME 2, and 20 U/mL of IL-2. Cells were seeded in a 24-well tissue culture plate at 1.5 mL aliquots per well and incubated for ten days in a humidified environment at 37 °C with 5% CO_2_. On day seven of the CoCx, 0.5 mL/well of old media was removed and 1 mL of fresh CTL media was added. On day ten, cells from all conditions were harvested and analyzed by flow cytometry and mass cytometry. A schematic of the experiment is depicted in Figure 1A.

### 2.5. Pentamer Quantitative Flow Cytometry

Cells from the DC-PBMC CoCx were analyzed by HLA pentamer quantitative flow cytometry to detect antigen-specific T-cells. Human HLA–B8 pentamer complexes with immunodominant peptide within full-length BZLF1 (RAKFKQLL (RAK)) conjugated with phycoerythrin (PE) (Proimmune, Oxford, UK) were used to detect and quantify BZLF1-specific CTL responses. Events were collected on Gallios (Beckman Coulter, Brea, CA, USA) or LSR Fortessa (BD Biosciences, Franklin Lakes, NJ, USA) flow cytometers and analyzed with Cytobank software (Beckman Coulter, Brea, CA, USA). Proportion and absolute event count of pentamer flow data of 4 donors was analyzed with linear mixed effects models. The absolute count of the cell populations was obtained by adding a known volume of microsphere counting beads (Thermo Fisher, Waltham, MA, USA) to a known volume of sample analyzed by pentamer flow cytometry.

### 2.6. T-Cell Repertoire of EBV-Specific T-Cells

BZLF1-specific T-cells (RAK+ T-cells) expanded in αDEC205-BZLF1 CoCx were sorted by fluorescence-activated cell sorting (FACS) to purity using an EBV pentamer. Sorted RAK+ T-cells (1 million cells per reaction) from 4 donors and their counterpart PBMCs, αDEC205-Ctrl CoCx product, and αDEC205-BZLF1CoCx product were subjected to quantitative analysis of their T-cell receptor (TCR) Vbeta (Vβ) repertoire by flow cytometry using the IOTest Beta Mark kit (Beckman Coulter, Brea, CA, USA). The IOTest Beta Mark kit contains monoclonal antibodies (mAbs) against 24 of the TCR-Vβ which covers approximately 70% of the normal human CD3 TCR Vβ repertoire populations. Results were compared to the reference normal range for TCR repertoire for peripheral blood which was computed by the OSU Department of Pathology, Laboratory of Flow Cytometry on 100 healthy volunteers and on data from the IOTest Beta Mark kit (Beckman Coulter, Brea, CA, USA).

### 2.7. In Vivo Preclinical Model of EBV-LPD

The Ohio State University Institutional Animal Care and Use Committee (OSU-IACUC) approved all animal work. Five to eight-week-old CB.17 scid/scid (SCID) mice were purchased from The Jackson Laboratory (Bar Harbor, ME, USA) and housed in a pathogen-free environment using our IACUC-approved protocol (2009A0094-R3-AR2). Animals showed no evidence of a leaky phenotype as determined by flow cytometry for T- and B-lymphocytes. Although SCID mice are lacking B- and T-cells, they do have functional NK-cells [35]. To enhance human PBMC engraftment, murine NK-cells were depleted using rabbit anti-asialo GM1 [36] anti-serum (Wako Chemicals, Richmond VA, USA). The GM1 anti-serum was injected by intraperitoneal (IP) route one day before PBMC injection (0.2 mg) and every week thereafter (0.1 mg). PBMCs from donor D-9 (HLA-B8/EBV+ donor) were injected via IP (5 × 10^7^ cells per mouse). Antigen-loaded (αDEC205-BZLF1 and αDEC205-Ctrl) DCs were injected into the mice (1.5–2 × 10^6^ cells/mouse) via IP at day 0, day 14, and day 28 post-transfer of human PBMC. To confirm human cell engraftment, ELISA was performed to quantify the amount of human IgG in mouse serum using a human IgG ELISA kit (eBioscience, San Diego, CA, USA) as per the manufacturer’s instructions. The IgG amount for two experimental groups was compared with a linear mixed model. Mice in the sentinel cohort (5 mice per group) were sacrificed 10 days after the last booster. Mice in the survival groups (10 mice per group) were monitored for signs of disease such as weight loss, ruffled fur, inactivity, and palpable abdominal masses. Necropsy and flow cytometry of mouse spleen was conducted to confirm EBV-LPD of all mice. Markers used for flow cytometry were: murine CD45, human CD45, human CD3, and human CD19.

### 2.8. ELISpot Assay

Splenocytes from mice (2 × 10^5^ cells per condition) in the sentinel cohort were stimulated with autologous lymphoblastoid cell line (LCL) which was incubated with BZLF1 pepmix (59 peptide pool of 15 mers with 11 amino acids overlap (JPT, Berlin, Germany). The splenocytes were also stimulated with BZLF1 pepmix alone, or anti-CD3 Ab. The quantity of stimulated splenocytes capable of secreting IFNγ was detected using the Human IFNγ Enzyme-Linked Immunosorbent Spot (ELISpot) kit as per manufacturer’s instructions (Mabtech, Nacka Strand, Sweden).

### 2.9. Mass Cytometry

Resting PBMCs, cells recovered from the CoCx, and murine splenocytes were prepared for mass cytometry (cytometry by time of flight, CyTOF) analysis as descried previously [37]. Briefly, 1 million cells were re-suspended in 50 µL of cell-staining buffer (CSF, Fluidigm, San Francisco, CA, USA), fixed with proteomic stabilizer (Smart Tube Inc., California-USA) for 10 min at room temper (RT) then stored at −80 °C until used. Fixed, frozen cells were thawed, washed once with cell-staining media (CSM) + 400 U/mL heparin, pelleted at 600 g for 5 min and incubated with Fc-blocker for 10 min at RT. The cells were stained with surface antibody (Ab) cocktail using a multi-parametric antibody panel (CyTOF Abs used provided in Appendix A). These Abs were either purchased or conjugated to heavy metals in-house using the Maxpar antibody conjugation kit as per manufacturer’s instructions (Fluidigm, San Francisco, CA, USA). Cells were incubated with the Ab cocktail for 50 min at RT with shaking and then washed three times with CSM. After washing, cells were permeabilized with 1 mL ice-cold methanol (MeOH) and incubated at −20 °C for 15 min. Following MeOH permeabilization, cells were stained with intracellular Abs (Appendix A) at RT for 50 min with shaking. Surface and intracellular Abs were either purchased or conjugated to the heavy metal in-house using the Maxpar antibody conjugation kit as per manufacturer’s’ instructions (Fluidigm, San Francisco, CA, USA). Subsequently, cells were washed once with CSM and once with phosphate-buffered saline (PBS), and then incubated with 125 nM iridium intercalator pentamethylcyclopentadienyl-Ir (III)-dipyridophenazine (Fluidigm, San Francisco, CA, USA) overnight at 4 °C. Prior to data acquisition, cells were washed once with CSM and twice with pure water and then re-suspended in pure water mixed with 1:20 (v:v) dilution of 4 elemental mass standard beads for normalization (Fluidigm, San Francisco, CA, USA) at 1 mL per million cells. Data were collected on a Helios mass cytometer (Fluidigm, San Francisco, CA, USA) at an event rate of 100 to 350 events per second. The following acquisition parameters were used: lower convolution threshold 600, event duration = 8 to 150, sigma = 3 along with noise reduction, randomization, and Gaussian discrimination. Bead events were removed using normalizer software [38]. Results were analyzed using Cytobank software (Beckman Coulter, Brea, CA, USA). Events were clustered using the unsupervised machine learning algorithms, viSNE and FlowSOM. The viSNE analysis is a visualization tool based on the t-distributed stochastic neighbor embedding (t-SNE) algorithm used to minimize the high-dimensional data in a two-dimensional map [39]. FlowSOM or flow unsupervised self-organizing map [40] is a clustering tool with self-organizing maps that show how all markers are behaving on all cells.

### 2.10. Cytotoxicity Assay

A flow cytometry-based cytotoxicity assay was utilized to measure cytotoxic activity of the cells collected after the DC-PBMC CoCx (effectors). Autologous LCLs (target cells) were pulsed with BZLF1 pepmix overnight, then washed and stained with cellTrace CFSE (Thermo Fisher, Waltham, MA, USA) for 30 min. Effector (E) and target (T) cells were incubated at specified E:T ratios for 4 h at 37 °C. The percentage of specific lysis was measured by flow cytometry as described previously [41].

### 2.11. Statistical Analyses

Linear mixed effects models were used for most of the experiments to take account of the correlation among observations from the same subject, as cells from the same set of donors were used under different treatment conditions, or the same animals were measured over time. Two sample t-tests were used for independent data, such as comparing immune response between the αDEC205-BZLF1 and the control group. Log-rank test was used for the survival analysis. Holm’s procedure was used to control for multiple comparisons when necessary. *p*-value < 0.05 after adjustment for multiple comparison is considered as significant.

## 3. Results

### 3.1. αDEC205-BZLF1 Fusion Protein Promotes Expansion of EBV-Specific T-Cells

To evaluate the ability of αDEC205-BZLF1 fusion protein to expand EBV-specific cytotoxic cells, we established CoCx from PMBC and autologous monocyte-derived DCs (Figure 1A). Ten days after CoCx set up, cells were analyzed by pentamer flow assay where we used an HLA-B8-restricted peptide (RAKFKQLL, henceforth referred to as RAK) to measure BZLF1-specific T-cells (RAK+ T-cells). The cell populations were identified by manual gating (Figure 1B), then T-cell populations (CD3+/linear negative; CD20-, CD14−, CD56−) were further analyzed by viSNE using six surface markers and RAK HLA-pentamer (Figure 1B). The viSNE immunome maps of CD3+ T-cells showed remarkable expansion of the CD3+CD8+RAK+ in αDEC205-BZLF1 CoCx (Figure 1C). The viSNE patterns were consistent across the four HLA-B8 healthy donors analyzed (Appendix A). Analysis with a linear mixed effect model confirmed that the frequency of RAK+ T-cells was significantly higher on αDEC205-BZLF1 CoCx cells compared to the PBMC baseline and cells produced in αDEC205-Ctrl CoCx with *p*-values of 0.0101 and 0.0091, respectively (Figure 1D). Similarly, absolute numbers of RAK+ T-cells were significantly higher in the αDEC205-BZLF1 CoCx (*p*-value of 0.016 for αDEC205-BZLF1 CoCx vs. PBMCs and *p*-value of 0.015 for αDEC205-BZLF1 CoCx vs. αDEC205-Ctrl CoCx) (Figure 1E). No significant difference in the frequency and absolute values of RAK+ T-cells between PBMCs and αDEC205-Ctrl CoCx was observed (*p*-value = 0.9297 and *p*-value = 0.159 respectively).

To analyze the differentiation phenotype of RAK+ T-cells, the intensities of individual immune marker expressions were visualized on viSNE maps. Tested markers included CD4, CD8, naïve, memory, and effector markers: CD45RO, CD45RA, CCR7, and CD62L, and RAK pentamer. The RAK+ T-cells were uniformly CD45RO+ (effector memory, (EM)) with few cells expressing both CD45RO and CD45RA, which illustrate a terminal effector (effector memory/CD45RA+ or EMRA) phenotype (Figure 2A). RAK+ T-cells showed heterogeneous CCR7 expression with even distribution of CCR7+ (Figure 2A, black arrow 1) and CCR7- (Figure 2A, black arrow 2) subsets. Interestingly, the residual RAK+ T-cells in undifferentiated donor-derived PBMCs were CD62L−, while the majority of the RAK+ T-cells expanded on the αDEC205-BZLF1 were CD62L+ (Figure 2A; black arrow 3), and fewer cells were CD62− (Figure 2A; black arrow 4). Very few RAK+ T-cells produced in the control CoCx were CD62+. We next generated biaxial plots colored according to the intensity of RAK+ pentamer which confirmed that RAK+ T-cells generated from the αDEC205-BZLF1 were mostly CD62L+ (Figure 2B). This pattern persisted across all four HLA-B8 donors, Appendix A). CD62L is a secondary lymphocyte homing receptor, and its expression implies that RAK+ T-cells expanded on αDEC205-BZLF1 are similar to a circulating effector phenotype. The expression of the lymphoid homing receptor enhances transition across high endothelial venules and entry into tonsils and in the oropharynx. Homing marker-positive T-cells (CD62L+/CCR7+) are more frequently associated with recognizing latent antigens but not lytic antigens [42]. Cells specific for the EBV lytic antigen, BZLF1, produced in αDEC205-BZLF1 CoCx included CD62L+/CCR7+ T-cells as well as CD62L−/CCR7− T-cells.

Expansion of both central memory (CM; CD45RO+/CD45RA−/CCR7+/CD62+) and effector memory (EM; CD45RO+/CD45RA−/CCR7−/CD62−) was significantly higher in the αDEC205-BZLF1 CoCx than in undifferentiated donor-derived PBMCs or cells produced in the control CoCx, as analyzed by a linear mixed effects model (Figure 2C).

### 3.2. Effector Cells Expanded in αDEC205-BZLF1 CoCx Exhibited Activation and Cytotoxic Phenotype and Show Enhanced Anti-Tumor Effects In Vitro

We next evaluated the immunophenotype of CoCx products by utilizing mass cytometry with a larger panel of antibodies (Appendix A). We executed immunophenotypic analysis of the data through manual gating (Figure 3A and Appendix A) and by automated viSNE clustering using surface and intracellular immune markers. An example of viSNE overlaid immunome maps of PBMCs, αDEC205-Ctrl, and αDEC205-BZLF1 from donor D-9 is displayed in Figure 3B. Both manual gating and the viSNE maps show distinct clusters representing different cell types. We observed expansion of CD8+ T-cells and contraction of CD4+ T-cell subsets in αDEC205-BZLF1 CoCx compared to αDEC205-Ctrl and PBMCs. We also observed an expansion of CD4/CD8 double-positive T-cells and CD4/CD8 double-negative T-cells on αDEC205-BZLF1 CoCx. Double-positive T-cells are frequently associated with control of viral infections such as HIV [43] and EBV [44]. Delayed recovery of double-negative T-cells is associated with EBV reactivation [45].

RAK pentamer was not included in the mass cytometry panel; however, we know from the pentamer flow assays that the majority of EBV-specific cells were identified within the CD3+CD8+CD45RO+CD62L+ gate (Figure 2B). Therefore, we generated an overlaid viSNE map of these cells (CD3+/CD8+/CD45RO+/CD62L+ effector cells) and total live/CD45+ cells (Figure 3C). The overlay maps confirmed that αDEC205-BZLF1 CoCx products demonstrated a significant expansion of CD3+CD8+CD45RO+CD62L+ effector cells, while the αDEC205-Ctrl did not (Figure 3C). Next, we evaluated the immunophenotype of this population across activation and cytokine markers. Biaxial plots colored according to the intensity of specific markers demonstrated that effector cells from αDEC205-BZLF1 CoCx were highly activated, expressing HLA-DR, ICOS, and Tbet. These cells also overexpressed the major recognition receptor, NKG2D, a c-type lectin-like receptor that enhances TCR signaling and aids in the recognition of targets [46]. We also observed upregulation of the degranulation surface markers CD107a and GRZB (Figure 3D) in T-cell populations from αDEC205-BZLF1 CoCx.

These observations strongly suggest that CTL cells produced in the αDEC205-BZLF1 CoCx can exert a cytotoxic function and could induce a highly effective immune clearance of EBV-infected B-cells. To evaluate the CTL functional activity, we performed a flow cytometry-based cytotoxicity assay using autologous EBV-immortalized LCLs pulsed with BZLF1 pepmix as target cells. Effectors and target cells were incubated for 4 h at specified effector:target (E:T) ratios. The results were analyzed with a linear mixed effect model (adjusted for multiple comparisons) and show that effector cells from αDEC205-BZLF1 CoCx have significantly higher killing activity both at 20:1 (*p*-value = 0.0163) and 40:1 (*p*-value = 0.0051) E:T ratio, compared to the αDEC205-Ctrl CoCx product (Figure 4).

### 3.3. Clonal Expansion of the TCR Repertoire of RAK+ T-Cells

To determine whether the RAK+ T-cells expanded in the αDEC205-BZLF1 CoCx are monoclonal, oligoclonal, or polyclonal in nature, we next assessed TCR Vβ repertoires. FACS-sorted RAK+ T-cells derived from the αDEC205-BZLF1 CoCx underwent an exploration of the TCR repertoire using flow cytometry. Quantitative analysis of RAK+ TCR Vβ showed that, among all donors tested, at least one of the Vβ 4, Vβ 7.1, or Vβ 14 were expanded to ≥1.5 of the upper limit of the reference normal range. The percent values corresponding to each RAK+ Vβ and the frequent moderate expansion across four donors recorded are illustrated in Figure 5A and Table 1. Remarkably, the TCR Vβ 4 represented close to 50% of the whole TCR repertoire of RAK+ T-cells from donor # D-9 (Table 1). The expansion of T-cells positive for beta families Vβ 4, Vβ 7.1, or Vβ 14, suggests the BZLF1-specific immune response is oligoclonal in nature.

Similar TCR Vβ distribution and expansion was also observed on T-cells within non-sorted αDEC205-BZLF1 CoCx (Figure 5B). No expansion of any particular TCR Vβ was observed in T-cells from PBMCs or αDEC205-Ctrl CoCx conditions, with the exception of Vβ 4 from αDEC205-Ctrl CoCx collected from donor # D-78.

### 3.4. Vaccination of Hu-PBL-SCID Mice with Anti-DEC205-BZLF1 Drives BZLF1-Specific Immunity and Improves Survival from Fatal EBV-LPD Disease

To assess the activity of αDEC205-BZLF1 in prevention of EBV-LPD, we utilized the Hu-PBL-SCID model. In this model, engraftment of PBMC from a seropositive EBV donor leads to the spontaneous development of a human lymphoproliferative disorder in the setting of profound human immune deficiency [47]. SCID mice were vaccinated with autologous mature DCs loaded with either the αDEC205-Ctrl or αDEC205-BZLF1 at day 0, day 14, and day 28 post PBMC transplant. Engraftment of human cells was assessed by ELISA measuring human IgG in mouse serum. Hu-PBL-SCID mice from both groups showed an increase in human IgG level over time (*p* < 0.0001) but no difference between experimental and control groups across all time points (*p* = 0.178) (Appendix A), consistent with equal engraftment between both treatment arms.

We designed this experiment to include survival and sentinel cohorts so that we could evaluate differences in immune responsiveness between the treatment groups. Mice in the sentinel cohort from each group were sacrificed 10 days following the last vaccination dose for evaluation of the immunophenotype of human mononuclear cell subsets with mass cytometry. Event files were imported into viSNE clustering algorithms by gating on human CD45 (HuCD45) cells. The majority of the human cells recovered from spleen were CD3+ T-cells (Figure 6A). There was no difference in frequency of lineage subsets or T-cell subsets between the two groups. To further define and quantify T-cells, we performed FlowSOM clustering of CD3+ T-cells from the two vaccination groups. The proportional node size of the FlowSOM minimal spanning tree (MST) and the meta-cluster across both HuCD4 and HuCD8 are depicted in Figure 6B. The FlowSOM analysis identified an increase in one cluster within the HuCD8 T-cells, meta-cluster 10 (Figure 6C). The meta-cluster 10, which was enriched in the αDEC205-BZLF1 vaccination group relative to the control group, captured activated CD8+ T-cells (HLA-DR+), expressing IFNγ, Tbet, CD45RO, and CD62L (Figure 6D). This expanded human memory effector population has the same phenotype as RAK+ T-cells (CD3+/CD8+/CD45RO+/CD62L+) (Figure 6D; red circle) expanded on the αDEC205-BZLF1 CoCx. No significant difference between groups was observed in clusters within the CD4+ T-helper population. Nevertheless, CD4+ T-helper cells from αDEC205-BZLF1 expressed the T-helper 1 phenotype (Tbet+ and IFNγ+) which might be supportive of lymphoma prevention and favorable outcome (Figure 6D).

To measure the immune responses among the sentinel cohort ex vivo, the number of IFNγ-secreting splenocytes was quantified by ELISpot. IFNγ was used as a surrogate marker of cell activation after stimulation with (1) autologous LCLs pulsed with BZLF1 pepmix, (2) BZLF1 pepmix alone, or (3) αCD3 antibody. Stimulation with autologous LCL pulsed with BZLF1 pepmix induced significantly a higher number of IFNγ-secreting cells than BZLF1 pepmix alone or αCD3 stimulation (*p* = 0.0002 and *p* < 0.0001, respectively, Figure 6E). Although the mean average of IFNγ-secreting cells stimulated with autologous LCLs pulsed with BZLF1 pepmix and pepmix alone was higher for the BZLF1-vaccinated group than the control group, the difference between groups was not significant. This reflects an inherent diversity of exposure and immune response to BZLF1 recall antigen [48].

There were 10 mice per group on the survival cohort. The log rank analysis showed survival was significantly improved in the αDEC205-BZLF1-vaccinated group (*p* = 0.046), remarkable for a highly immunosuppressed environment (Figure 6F).

## 4. Discussion

Over 39,000 solid organ transplants were performed in the US in 2020 [49], and the number of transplants continues to increase each year. PTLD often occurs within the first year following transplantation [12] and is linked with a high mortality and morbidity due to allograft loss [50]. The use of potent, multi-drug IS therapy to prevent graft rejection places up to 20% of transplant recipients at risk for developing PTLD [51,52,53,54]. The majority of PTLD is associated with EBV infection [55] in the setting of heavy iatrogenic IS that impairs the development of critical cellular adaptive immune responses [56,57]. EBV-seronegative patients are particularly vulnerable to the development of PTLD [58] and represent a group of patients that could be ideal candidates for preventive vaccination strategies to protect from primary EBV infection that invariably occurs in the post-transplant setting after receipt of an organ from EBV-positive donors. Non-vaccine-based methods to reduce the risk of PTLD have been reported, including use of EBV-seronegative organ donors, prophylactic treatment with anti-viral medications [59,60], use of rituximab as an induction regimen [59] and upon onset of EBV viremia [61], and upfront use of EBV-specific cytotoxic T-cell (CTL) products [62] in patients believed to be at high risk of PTLD. The majority of these studies are either retrospective in nature, involve low numbers of patients, or report results that contradict outcomes of similar trials. The proposed use of organ donors who are seronegative, and pre-emptive use of EBV-specific CTLs, while likely to be effective, are impractical and unable to be utilized at most centers.

Successful development of a vaccine to prevent EBV infection or EBV-associated diseases remains an unmet need [25]. To date, the majority of EBV vaccine efforts have focused on the viral envelope glycoprotein 350 (gp350) as a vaccine immunogen [63]. The gp350 binds to the CD21 complement receptor on B lymphocytes and facilitates entry of the virus into host cells [64]. Vaccination of cotton top tamarins with purified gp350 protected from EBV-LPD but did not prevent primary infection [65]. The first effort to test an EBV gp350 target protein in humans utilized a live recombinant vaccinia virus expressing the EBV antigen under the 11 K vaccinia promoter [66]. This phase I clinical trial vaccinated children and adults and led to increased antibody titers to gp350, but did not prevent infection with EBV. Recombinant gp350 vaccine produced in Chinese hamster ovary cells was tested in a double-blinded randomized phase I clinical trial [67]. EBV-seropositive and -seronegative adults were vaccinated and were shown to produce neutralizing antibodies to gp350, most efficiently when alum/monophosphoryl lipid A (alum/MPL) adjuvant was co-administered [67]. Another trial (phase II clinical trial) tested the delivery of recombinant gp350 (50 μg dose delivered three times at 0, 1, and 5 months) with alum/MPL in EBV-seronegative adults [68]. The 180 normal volunteers in the study were randomized to receive vaccine or placebo control (90 per group). Follow up was for 18 months and symptoms of infectious mononucleosis were monitored. One month after the final dose of vaccine, 98.7% of subjects had positive anti-gp350 antibody titers; however, the incidence of symptomatic EBV infection was similar in both groups. Intention to treat analysis showed a significant difference in vaccine efficacy to prevent infectious mononucleosis (78% risk reduction, *p*-value = 0.03). The first phase I trial administered to EBV-seronegative pediatric patients awaiting solid organ transplantation tested two doses (12 and 25 μg) of recombinant gp350 with alum adjuvant [69]. Of 13 patients that were vaccinated, only 4/13 individuals developed neutralizing antibodies, and 2/13 became infected with EBV during follow up. A small but promising study focusing on EBNA3A reported a randomized, single-blinded, placebo-controlled phase I study in HLA-B*08:01, EBV-seronegative young adults [70]. This study, delivering two doses (5 and 50 μg) of EBNA3A peptide with tetanus toxoid in water oil emulsion (montanide ISA720), led to memory T-cell responses in eight of nine vaccinated individuals. None of the vaccinated subjects developed infectious mononucleosis compared to one of four subjects treated with placebo. These results, while promising, were not adequately powered for statistical evaluation. Collectively, the majority of vaccine strategies tested in preclinical and clinical trials focused on the humoral response with little attention directed toward the adaptive cellular response.

Adaptive cellular immunity specific for EBV has been shown to have important prophylactic and therapeutic activity in several EBV-associated malignancies, including PTLD [71,72,73,74]. Donor EBV-specific T-cells have been used to prevent and treat EBV-LPD in patients undergoing allogeneic HSC transplantation. EBV-specific T-cells were generated by repeated in-vitro sensitization with autologous LCL in the presence of IL-2. Donor-derived CTLs were infused into 36 allogeneic HSC transplant patients. The prophylactic treatment decreased the EBV viral load at 2–4 log from study entry level and none of the patients developed PTLD. Infusion of EBV-specific CTLs to control patients who developed PTLD led to complete response and full recovery [75]. Bollard et al. used EBV-specific T-cells to treat patients with relapsed Hodgkin’s lymphoma (HL). EBV+ HL typically expresses a latency II pattern of EBV gene products, including EBNA1, LMP1, and LMP2. Adoptive transfer of patient-derived EBV-CTLs for therapy of relapsed HL demonstrated a two log-fold, in vivo expansion of the T-cell product in peripheral blood and reduction in the viral DNA levels. Clinically, EBV-CTL infusions led to complete remission in five patients for more than 40 months [18]. Despite the success of donor- and patient-derived EBV-specific CTLs, development of these cellular products is time and labor intensive.

Off-the-shelf T-cells from third-party donors provide immediate access to treatment. Prockop et al. reported results of a phase II clinical trial using EBV-specific CTLs derived from third-party donors. Forty six patients with rituximab refractory lymphoma after allogenic HCT (*n* = 33) or SOT (*n* = 13) were treated with three weekly infusions of third-party, partially HLA-matched EBV-specific CTLs. The CTL infusion was well tolerated, with no evidence of toxicity except for one patient who developed a transient skin rash that resolved with topical steroid therapy. Clinically, 68% of HSCT and 54% of SOT recipients achieved a complete response (CR) or durable partial response (PR) [76].

Prior work by our group has shown BZLF1-specific CTL expansion, following reduction in immune suppression, correlated with PTLD tumor regression and improved survival [26]. The study followed a standardized management approach to patients with PTLD after renal transplantation. The protocol involved a uniform strategy to reduce IS medication, discontinuation of one IS drug while continuing cyclosporine (50% dose reduction), and steroid taper (to target 5–10 mg daily). Median follow up was for >120 months with close monitoring of renal function, EBV viral load, and, when possible, antigen-specific CTL responses using viral peptide-loaded tetramers and flow cytometry. Ninety one percent of the patients achieved CR and remained disease free. Renal transplant rejection occurred in five (45%) patients, and the four patients with graft loss had PTLD involving the transplanted organ. Serial PBMC samples from two HLA-B8 patients were analyzed by flow cytometry with MHC/peptide tetramers. Robust endogenous responses to BZLF1 (RAK+ T-cells) correlated with tumor regression and survival [26].

Our data in the current study display that the αDEC205-BZLF1 vaccine platform elicited strong EBV-specific, effector memory cellular responses in vitro and in vivo. The αDEC205-BZLF1 fusion product led to significant expansion of EBV-specific central and effector memory CTLs (CM, EM). RAK+ CM cells express high levels of the homing marker CD62L, which allows for migration to secondary lymphoid organs for rapid respond to recall antigens. αDEC205-BZLF1 led to expansion of EM cells which have an immediate effector function against targets. CM and EM T-cells are crucial to provide long-term protective immunity against EBV [77]. Effector cells produced in the CoCx with DCs loaded with αDEC205-BZLF1 were activated (HLA-DR+/ICOS+), and expressed high levels of Tbet, NKG2D, CD107A, and GRZB (Figure 3D). These observations strongly confirm that CTLs generated in the αDEC205-BZLF1 CoCx possess cytotoxic function and would likely be effective in immune surveillance of EBV-infected B-cells. Indeed, cells from αDEC205-BZLF1 CoCx demonstrated potent cytotoxicity against autologous EBV + LCL targets in vitro (Figure 4). Spleen cells from mice in the αDEC205-BZLF1 vaccination group responded to recall antigen (autologous LCL pulse with BZLF1) by increasing the expression of IFNγ (Figure 6E). The improved survival seen in the Hu-PBL-SCID model (Figure 6F) is remarkable given the significant immune dysfunction that has been reported. Such activity justifies further examination in clinical trials.

Other groups have utilized the αDEC205 platform to examine strategies to target EBV-associated disease. Gurer et al. tested the αDEC205 fusion protein platform for vaccine development [78] using the EBV latent antigen EBNA1 as an immunogen. αDEC205-EBNA1 fusion protein vaccination elicited EBNA1-specific CD4+ Th memory responses in humanized murine models, driving expansion of EBNA1-specific IFNγ secreting T-cells. The EBNA1 protein is characterized by a Gly-Ala repeat that reduces the efficiency of intracellular protein processing and endogenous presentation by HLA molecules and is poorly recognized by CD8+ T. Nevertheless, use of αDEC205-EBNA1 vaccine led to the expansion of EBNA1-specific memory CTL cells.

We believe that conditions for the development of a successful vaccination approach to prevent PTLD using BZLF1 as a target antigen can be satisfied. First, the population at high risk can be identified (EBV-seronegative patients awaiting solid organ transplantation). Given this, it is possible to vaccinate patients prior to undergoing solid organ transplantation when they are fully immune-competent. While patients with compromised organ function (especially end-stage renal disease) are relatively immunocompromised, there is ample evidence supporting vaccination of such patients to prevent disease [79]. Second, through work accomplished in our laboratory, we now have in vivo evidence from both our animal model [28] and our successfully treated patients diagnosed with EBV+ PTLD [26] as to which EBV antigens are seen by T-cells when a successful anti-tumor immune response is mounted. Surprisingly, we identified a correlation between the emergence of a robust BZLF1-specific memory CD3+CD8+ T-cell response, PTLD regression, and survival. Third, BZLF1 antigen is viral and therefore completely foreign to humans, eliminating concerns about specificity and tolerance. Fourth, we have multiple biomarkers to quantify the T-cell response to this antigen that, in turn, correlate with prevention or resolution of EBV-LPD. Finally, we have what we believe to be an analogous chimeric mouse–human model of EBV-LPD, in which SCID mice are reconstituted with components of a human immune system and, with certain donors seropositive for EBV, spontaneously develop human EBV-LPD that is highly analogous to PTLD [80,81,82]. Indeed, we have reported that several viral antigens (EBNA3a, BZLF1) [83] recognized by the human T-cells in the mouse model are the same antigens recognized by T-cells in PTLD patients [83]. Furthermore, patients that develop PTLD in the first year post transplant showed elevated levels of BZLF1 protein detected in peripheral blood [84]. While we believe this collective data supports the use of BZLF1 as a vaccine immunogen for exploration in our first clinical trial, it remains plausible that other latent (LMP1, LMP2, EBNA2, EBNA3a, b, c) gene products will be required for a fully comprehensive vaccine to protect against EBV-driven disease. We are currently investigating other vaccine platforms to deliver full-length viral proteins for use in the Hu-PBL-SCID model of EBV-LPD and anticipate future clinical trials exploring the use of such target proteins.

Our results from this BZLF1 immunization study provide a rationale for testing a BZLF1 vaccine in clinical trials. Work extended from this finding may potentially be relevant to other immunocompromised individuals at risk of EBV-associated malignancies, including patients with HIV+ infection and individuals with primary immune deficiency disorders. Furthermore, EBV is associated with a wide variety of malignancies in immunocompetent patients globally, including lymphomas (Burkitt lymphoma, classical Hodgkin’s lymphoma, and aggressive diffuse large B-cell and extranodal NK-cell lymphomas [85,86]) and nasopharyngeal and gastric carcinomas [87,88]. The global burden of EBV-driven cancers has recently been estimated at 18% of all cancers, with approximately 164,000 deaths in 2017 alone [89], and an increased incidence and mortality of 36 and 19%, respectively, since 1990. Thus, vaccination strategies to prevent EBV-associated malignancies in immune-competent individuals represents an attractive approach to impact cancer prevention on a global scale.

## Figures and Tables

**Figure 1 vaccines-09-00555-f001:**
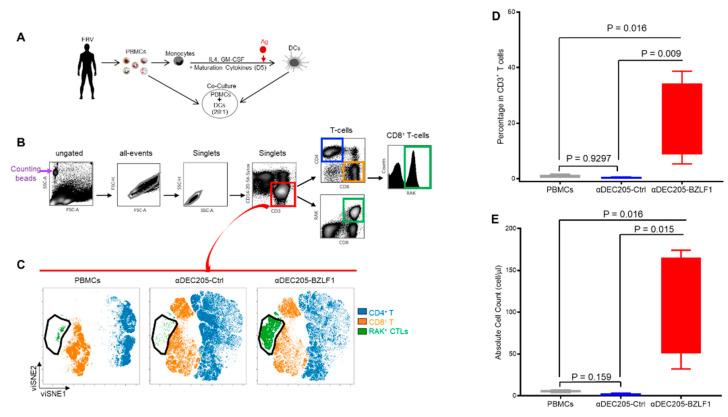
In vitro evaluation of BZLF1 immunogen through DC: PBMC CoCx, followed by HLA-pentamer loaded with BZLF1 RAK peptide and flow cytometry; (**A**) Schematic of the CoCx experiments; (**B**) Gating scheme for pentamer flow data; (**C**) viSNE immunome maps that show distinct clusters representing different immune cell types; (**D**) Frequency of RAK+ T-cells from PBMCs, αDEC205-Ctrl CoCx, and αDEC205-BZLF CoCx across 4 HLA-B8 donors; (**E**) Absolute count of RAK+ T-cells from PBMCs, αDEC205-Ctrl CoCx, and αDEC205-BZLF CoCx across 4 HLA-B8 donors.

**Figure 2 vaccines-09-00555-f002:**
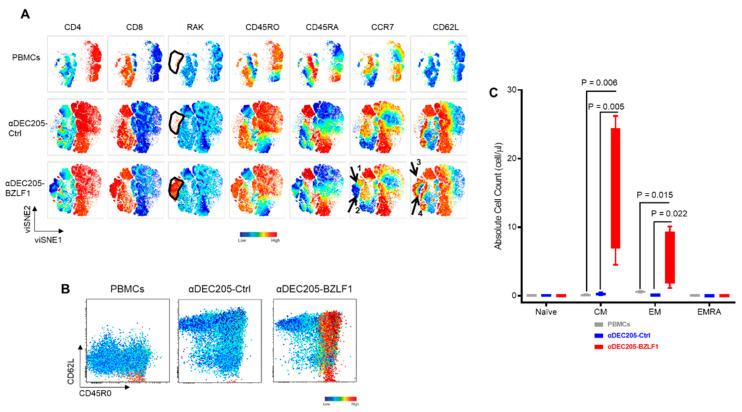
Characteristics of EBV-specific cells in the CoCx: (**A**) The viSNE maps, colored by the intensities of individual immune marker expression (CD4, CD8, RAK, CD45RO, CD45RA, CCR7, and CD62). For each marker tested, cells were separated according to the expression of particular marker. RAK+ memory T-cells generate on the αDEC205-BZLF1 CoCx were evenly distributed between the CCR7+ (black arrow 1) and CCR7− (black arrow 2). Majority of the RAK+ cells expanded on the αDEC205-BZLF1 are CD62L+ (black arrow 3) while fewer were CD62L− (black arrow 4); (**B**) Biaxial plots of CD8+ T-cells with CD45RO on the *X*-axis versus CD62L on the *Y*-axis. Events on the biaxial plots colored according to the RAK intensity expression which confirmed that the memory EBV-specific cells from the PBMCs and αDEC205-Ctrl lack the lymphocyte homing marker, CD62L, while the majority of the RAK+ cells generated from the αDEC205-BZLF1 are CD62L+; (**C**) RAK+ naïve and memory subsets calculated from PBMCs, αDEC205-Ctrl CoCx, and αDEC205-BZLF CoCx across 4 HLA-B8 donors.

**Figure 3 vaccines-09-00555-f003:**
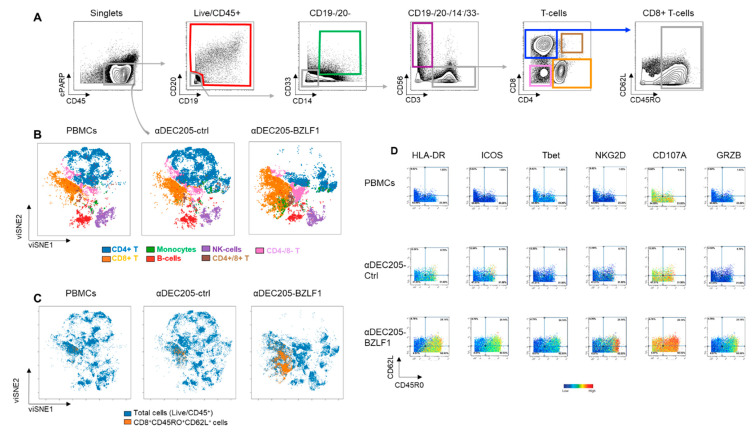
Immunome phenotype of cells generated on the DC: PBMC CoCx, by mass cytometry: (**A**) Gating scheme; (**B**) viSNE immunome maps that show distinct clusters representing different immune cell types; (**C**) Overlaid viSNE maps of total cells and CD8+CD45+CD62L+ effector cells; (**D**) Biaxial plots of CD8+ T-cells with CD45RO on the *X*-axis versus CD62L on the *Y*-axis. Evens on the biaxial plots colored according to the intensity of expression of the indicated markers.

**Figure 4 vaccines-09-00555-f004:**
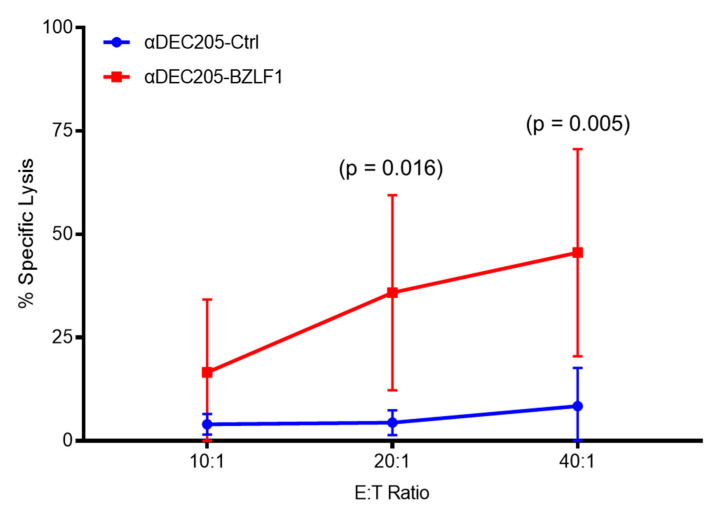
Percentage of killing on flow cytometry-based cytotoxicity assay at three E:T ratios.

**Figure 5 vaccines-09-00555-f005:**
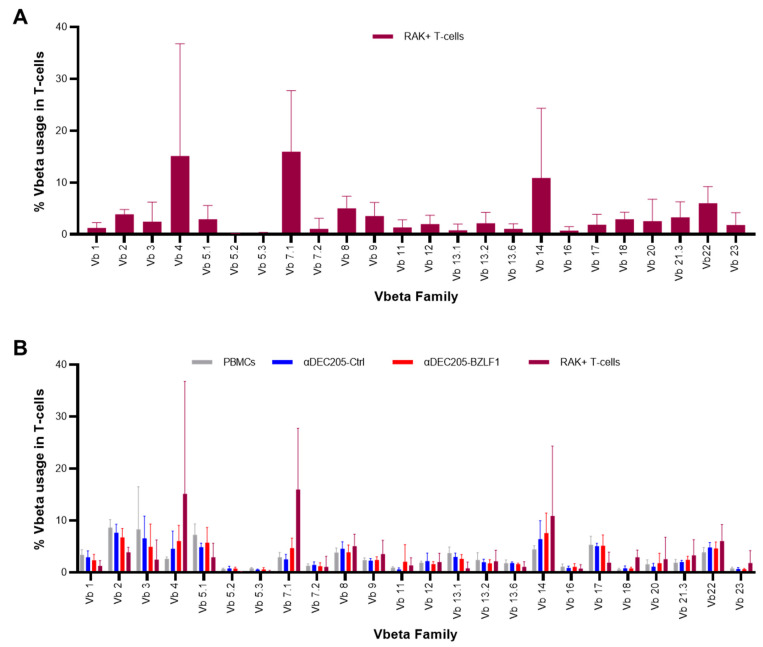
TCR clonogram: (**A**) TCR clonogram of Vβ repertoire in RAK+ T-cells across 4 HLA-B8 donors; (**B**) Comparison of the TCR-Vβ repertoire between PBMCs, αDEC205-Ctrl CoCx, αDEC205-BZLF CoCx, and RAK+ T-cells across 4 HLA-B8 donors.

**Figure 6 vaccines-09-00555-f006:**
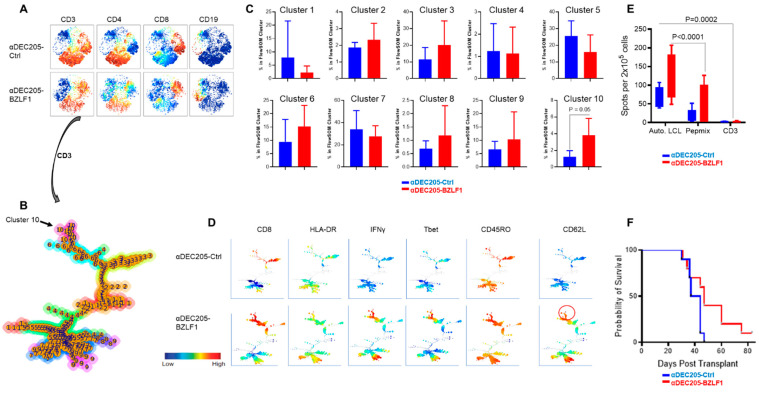
In vivo evaluation of the αDEC205-BZLF immunogen; (**A**) The viSNE maps, colored by the intensities of individual immune marker expression (CD3, CD4, CD8, and CD19); (**B**) FlowSOM minimal spanning tree (MST) on HuCD3+ T-cells showed both HuCD4+ T-cells and HuCD8+ T-cells; (**C**) Percentage of T-cell population from each vaccination group (αDEC205-Ctrl CoCx and αDEC205-BZLF) in each FlowSOM meta-cluster; (**D**) FlowSOM minimal spanning tree (MST) colored by the intensities of individual immune marker expression (CD8, HLA-DR, IFNγ, Tbet, CD45RO, and CD62); (**E**) Ex-vivo immune responsiveness to recall antigens (autologous LCLs, BZLF1 pepmix, and anti-TCR) by ELISpot; (**F**) Log-rank test: BZLF vaccine significantly delayed lymphoma development.

**Table 1 vaccines-09-00555-t001:** Distribution of 24 TCR-Vβ families in RAK+ T-cells from 4 HLA-B8 donors.

TCR	Normal Range	EBV-Specific CTLs			
		D-9	D-77	D-78	D-81
Vb 1	1.89–11.7	0.6	2.8	0.2	0.97
Vb 2	4.03–23.48	4.4	6.2	2.4	2.3
Vb 3	0.52–15.71	0.6	7.6	0.1	0.6
Vb 4	0.79–3.26	47.5	2.2	6.4	0.8
Vb 5.1	3.19–14.93	0.4	1.5	2.3	4.9
Vb 5.2	0.49–4.98	0.1	0	0.2	0.1
Vb 5.3	0.37–2.98	0.1	0	0.3	0.4
Vb 7.1	0.64–20.01	6.5	21.7	5.1	26.1
Vb 7.2	0.05–5.45	0.1	0	0	8.8
Vb 8	2.26–29.47	4.1	5.5	2.3	6.4
Vb 9	1.1–9.3	1.9	1.8	3	7
Vb 11	0.25–5.11	1.3	0	0.7	0.3
Vb 12	1–4.76	1.9	0.3	0.6	4.4
Vb 13.1	1.62–8.16	0.1	0.1	2.6	0.3
Vb 13.2	0.80–5.28	0.1	2	1.3	1.5
Vb 13.6	0.84–8.8	0.2	1	0.4	1.3
Vb 14	1.33–8.03	0.4	25.6	0.1	12.7
Vb 16	0.42–1.9	0.7	0.3	0.1	0.5
Vb 17	2.28–12.61	0.6	1	0.6	2.2
Vb 18	0.58–5.23	3	4.5	2.2	1.4
Vb 20	0–9.73	0.2	2.2	0.1	5.4
Vb 21.3	1.08–5.97	1.8	7.6	1.6	1.9
Vb22	1.99–9.89	9.1	2	5.4	7.2
Vb 23	0.26–4.76	0.4	1	0.6	1.8

Clonal (>2× of the higest end on the noral range); Oligoclonal (≥2× of the highest end on the noramal range); Expanded (≥1.5× of the higest end on the normal range); Within normal range.

## Data Availability

All data pertaining to this study are contained within the article.

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
