# Peer review of "Targeted Delivery of BZLF1 to DEC205 Drives EBV-Protective Immunity in a Spontaneous Model of EBV-Driven Lymphoproliferative Disease"

_vaccines, 2021, doi:10.3390/vaccines9060555_

Round 1

Reviewer 1 Report

The Authors  have developed a vaccine to bolster EBV-specific immunity to the BZLF1 protein and show that co-culture of dendritic cells (DCs) loaded with a αDEC205-BZLF1 fusion protein with peripheral blood mononuclear cells (PMBCs) leads to expansion and increased cytotoxic activity of central-effector memory CTLs against EBV-transformed B-cells.

These are very interesting results as EBV vaccine may be useful also to prevent other EBV Associated Malignancies.The Authors shoud comment on this possibilty.

Authors should comment on the possibility to prevent other EBV Associated Malignancies as there is recent evidence  that the spectrum of EBV associated lymphomas may be wider than currently known  (Mundo et al. Modern Pathology 2020)

Reviewer 2 Report

Ahmed and colleagues present results of a vaccine study against EBV with the aim of preventing PTLD after solid organ transplant. The rationale for this study is based on the group's prior work showing a correlation between PTLD regression and BZLF1 specific CTL memory expansion. PBMCs from EBV+ HLA-B8 donors were used to generate dendritic cells, which were co-cultured with aDEC205-BZLF1 fusion protein.. The result was clonal expansion of EBV-specific T cells including central effector and memory T cells in vitro. Vaccination in SCID mice drove BZLF1 specific immunity and improved survival. The investigators propose that this study lays down the framework to move forward with testing of BZLF1 vaccines in clinical trials. 

The study is thoughtful and the investigators are correct that PTLD is associated with a very poor prognosis and preventative strategies are desperately needed. Therefore this is an important study. However, they propose vaccinating EBV seronegative transplant recipients, which are mainly limited to pediatric or young adult patients (since EBV is ubiquitous and infects 90% of the worlds population by late adulthood). Therefore this study has the potential to impact a narrow pool of solid organ transplant recipients. The investigators should also acknowledge that success of this vaccine in humans would require the patient's immune system to mount a response to the vaccine, which might be difficult in the setting of immunosuppression therapy or in a patient with multiple comorbidities rendering them immunocompromised (such as end stage kidney disease). In the discussion, it would strengthen the manuscript if the authors would compare their BZLF1 vaccination with other preventative strategies such as prophylactic rituximab or EBV CTLs or of course attempting to utilize EBV seronegative grafts if possible. Lastly, it would strengthen the manuscript if the investigators would comment a bit more on why the BZLF1 immunogen is a better vaccine antigen than type III latency proteins which are very immunogenic. 

Reviewer 3 Report

This study by Ahmed et al titled “ Targeted Delivery of BZLF1 to DEC205 Drives EBV-Protective Immunity in a Spontaneous Model of EBV-Driven Lymphoproliferative Disease” studied using BZLF1 as an immunogen to harness adaptive cellular responses and prevent PTLD. This is a well done study with appropriate research design.
